# Study and Mechanism Analysis on Dynamic Shrinkage of Bottom Sediments in Salt Cavern Gas Storage

**Baocheng Wu [1], Mengchuan Zhang [2], Weibing Deng [1], Junren Que [1], Wei Liu [1], Fujian Zhou [2], Qing Wang [2], Yuan Li [2] and Tianbo Liang [2,*]**

[1] CNPC Engineering Technology Research Institute of Xinjiang Oilfield Company, Karamay 834000, China; wubc@petrochina.com.cn (B.W.); dengweibing@petrochina.com.cn (W.D.); quejunren@petrochina.com.cn (J.Q.); liuwei2006@petrochina.com.cn (W.L.)

[2] State Key Laboratory of Oil and Gas Resources and Prospecting, China University of Petroleum at Beijing, Beijing 102249, China; mcmengchuan@163.com (M.Z.); zhoufj@cup.edu.cn (F.Z.); 18811731154@163.com (Q.W.); 2017310210@student.cup.edu.cn (Y.L.)

[*] Correspondence: liangtianboo@163.com

**Abstract:** Underground salt cavern gas storage is the best choice for the production peak adjustment and storage of natural gas, and is a basic means to ensure the safe supply of natural gas. However, in the process of these caverns dissolving due to water injection, argillaceous insoluble sediments in the salt layer will fall to the bottom of the cavity and expand, occupying a large amount of the storage capacity and resulting in the reduction of the actual gas storage space. Effectively reducing the volume of sediments at the bottom of the cavity is a potential way to expand the storage capacity of the cavity. In this study, a method to reduce the volume of argillaceous insoluble sediments with particle sizes ranging from 10 mesh to 140 mesh, via a chemical shrinkage agent, has been proposed. Firstly, the inorganic polymer shrinkage agent PAC30 was synthesized, and then a set of dynamic shrinkage evaluation methods was established to evaluate the influence of temperature, particle size, concentration, and other factors on the shrinkage performance. Finally, by means of a scanning electron microscope (SEM), the Zeta potential, and static adsorption experiments, the mechanism of the interaction between PAC30 and cavity-bottom sediments was described and verified in detail. The experimental results show that the optimal concentration of PAC30 for dynamic shrinkage is 20 ppm. The shrinkage performance of PAC30 decreases with an increase in temperature, and the smaller the particle size of the insoluble sediments, the worse the shrinkage performance. According to the adsorption experiment and Zeta potential, PAC30 can be effectively adsorbed on the surface of insoluble sediments, and the SEM images show that, after adding PAC30, the particles are tightly packed, and the volume of insoluble sediments is significantly reduced. In the large-scale model experiment, the expansion rate of PAC30 reached 20%, which proves that the shrinkage agent is a potential method to expand the gas storage volume.

**Keywords:** shrinkage agent; underground salt cavern gas storage; insoluble sediments; dynamic shrinkage experiment

## 1. Introduction

Underground energy storage (e.g., natural gas, oil, H$_2$) is one of the most important topics for the sustainable development of the national economy [1]. Underground salt cavern gas storage is the best choice for the peak shaving of natural gas production and stocking up of natural gas resources and is the most basic means to ensure the safe supply of natural gas [2]. Gas storage refers to underground gas banks. Although extensive in the construction period, salt cavern gas storage is relatively stable in performance and allows the fast injection and quick recovery of gas. Traditional gas storage built with depleted sandstone gas reservoirs can only realize one cycle of gas injection in summer and gas recovery in winter every year. Traditional gas storage is similar to growing rice in one

season per year, while salt cavern gas storage is similar to growing rice in more than one season a year, which can meet the demand of peak adjustment in more diversified ways [3].

Salt layers in China have the characteristics of a dense structure, low porosity, low permeability, and self-healing ability, making them ideal media for oil and gas storage [4]. However, there are still some problems in building gas storage in salt layers. Salt cavern gas storage in China is characterized by multiple interlayers and high contents of insoluble sediments. For example, Jintan and Pingdingshan salt cavern gas storage facilities have insoluble sediments occupying 33–66.2% (on average 46.8%) of the total volume of the salt caverns, leaving limited cavity space for gas storage [5]. This is a problem worthy of attention [6].

Many scholars have studied insoluble sediments in gas storage. Li [7] studied the distribution characteristics of insoluble sediments in gas storage and proved that particle size was in a linear relationship with the volume of insoluble sediments. Sun [8] analyzed the factors affecting the sedimentation rate of insoluble residues. The results showed that the settling velocity of insoluble sediments was mainly affected by its own particle size and brine Bome, whereby the settling velocity of particles increased with the increase in particle size and decreased with the increase in brine Bome. Gan [9] investigated the expansion rates of insoluble sediments under different calcium sulfate mass fractions, temperatures, and particle sizes. They found that the expansion rate of insoluble sediments was negatively correlated with the mass fraction of calcium sulfate, and positively correlated with the temperature and particle size of insoluble sediments. Ren [10] measured the void volume of insoluble sediments using the gas expansion method and carried out an experimental study and analysis of the air injection and halogen discharge of insoluble sediments based on a physical modeling device. Chen [11] explored the influences of the shape, size, arrangement, and particle composition of broken rocks on the expansion coefficient using ideal experiments and logical analysis. They concluded that the reasons the volume of insoluble sediments at the bottom was larger than that in the interlayer were that salt rock became larger in volume when crushed and clay minerals absorbed water and expanded.

However, due to the late start of the study on the volume expansion of insoluble sediments in gas storage, there are few research and application examples in the oil field [12]. Moreover, most of the current studies focus on the sedimentation rate and accumulation morphology of insoluble sediments in gas storage [13–16], and there are few reports on reducing the volume of insoluble sediments. Investigating how to reduce the volume of insoluble sediments via the chemical method according to the existing experiences and rules remains to be studied.

In this study, in view of the expansion of insoluble sediments in salt cavern gas storage, a type of shrinkage agent suitable for insoluble sediment particles of 10–140 mesh at the bottom of gas storage was synthesized. Through a series of evaluation experiments, the shrinkage effect of the shrinkage agent was explored, and the interaction mechanisms between the shrinkage agent and insoluble sediments were discussed. The experimental results show that the shrinkage agent has the best shrinkage effect at the concentration of 20 ppm and can expand the effective volume of gas storage.

## 2. Materials and Methods

### 2.1. Materials

Sodium chloride (NaCl, 99%), aluminum chloride ($AlCl_3$, 99%), and potassium hydroxide (KOH, 99%) were purchased from China National Pharmaceutical Group. The insoluble sediments were taken from gas storage in Chuzhou, China.

### 2.2. Preparation of Shrinkage Agent PAC30

The shrinkage agent prehydrolyzed polyaluminum is the most widely used shrinkage agent and has been extensively studied. Compared with traditional $AlCl_3$, the polyaluminum shrinkage agent has a higher charge neutralization ability and faster aggregation speed [4]. It is generally believed that $Al_{13}$, with its strong structural stability, high charge

neutralization capacity, and nanoscale molecular diameter, is the core component of the polyaluminum shrinkage agent. However, in order to treat the insoluble sediments in gas storage better, its charge neutralization ability and molecular diameter need to be further optimized.

$Al_{30}([Al_{30}O_8(OH)_{56}(H_2O)_{24}]^{18+})$ polymer (PAC30) is a polycation with a Keggin structure in a hydrolyzed polyaluminum solution [17]. $Al_{30}$ is composed of two δ-$Al_{13}$ connected by four Al monomers. The two-tetrahedral coordinated Al in $Al_{30}$ produces wide [27]Al NMR signals at d = 70 ppm. $Al_{30}$ has a higher resistance to high temperatures and stronger charge neutralization ability than $Al_{13}$, and it has 18 positive charges and a unique nanomolecular size, making it another promising shrinkage agent besides $Al_{13}$ [18].

PAC30 was synthesized according to the method of Chen (2006). Specifically, 0.7 mol/L KOH was slowly added to 100 mL of a 1 mol/L $AlCl_3$ aqueous solution and the mixture was stirred at 60 °C until the alkalinity reached 2.4. Sufficient shear force was needed in the mixing process to prevent the formation of amorphous products. Then the solution was heated at 90 °C for 24 h under the conditions of stirring and condensation reflux. Then it was aged for five days. The final product was PAC30.

### 2.3. XRD Analysis of Insoluble Sediments

The mineral analysis was carried out by a whole-rock X-ray diffractometer (XRD) in three steps: The first step was to crush the rock powder to 400 mesh; the second step was to put the sample into the sample chamber and press the fixed-spring plate; the third step was to turn on the instrument for testing and analysis.

### 2.4. Establish a Shrinkage Agent Evaluation Method

At present, there is no standard test method for the evaluation of the shrinkage agent. According to the field construction process, this paper designed a set of evaluation methods for the shrinkage agent:

(1) Dynamic dissolution shrinking experiment.

Salt cavern gas storage in China is built by layered salt with a high content of insoluble sediments (15~40%), a thin salt layer (approximately 150 m), and a thick interlayer (3~11 m). After completion, a large amount of argillaceous insoluble matter will have accumulated in the cavity. In order to verify the shrinkage effect of the synthesized PAC30 and explore the influence of different factors on the shrinkage effect, the dynamic corrosion shrinkage and expansion experiment was established, and the experimental device was set up according to Figure 1. The experimental steps were designed as follows:

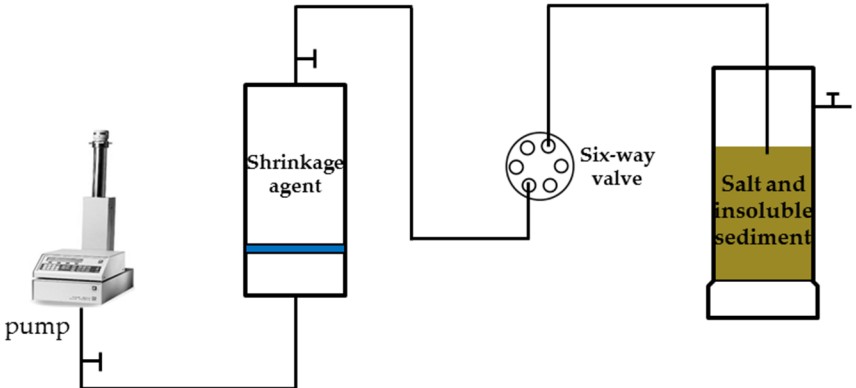

**Figure 1.** Schematic diagram of dynamic shrinkage experiment device.

Firstly, the insoluble sediments from the gas storage were ground into powder of corresponding meshes, washed with a dichloromethane solution to remove oil and salt for 24 h, and dried in an oven at 105 °C for 24 h until a constant weight was reached.

Secondly, a total of 200 g of insoluble sediment and NaCl were taken. The insoluble sediment: NaCl was evenly mixed at a mass ratio of 30:70 and then poured into a three-mouth flask. We then dropped different shrinkage agents or distilled water into the three-port flask at a constant flow rate of 0.2 mL/min and recorded the volume of the remaining solid every time for 168 h. In the range where the difference between the results was less than 5%, all the results were taken as the arithmetic average of the two groups. The calculation formula of the shrinkage rate is [19]:

$$K_1 = \frac{V_1 - V_2}{V_1} \times 100\% \tag{1}$$

where $K_1$ is the shrinking rate (%), $V_2$ is the expansion volume of insoluble sediments in the shrinkage agent solution (mL), and $V_1$ is the expansion volume of insoluble sediments in water (mL).

(2) Large-scale Dynamic Salt Solution Simulation.

The experimental device is shown in Figure 2. Figure 2a,b shows the self-built device, which is a cuboid container of 24 cm × 28 cm × 40 cm for storing insoluble sediments. Figure 2c shows the large-scale dynamic salt solution simulation device consisting of a shrinkage agent, a circulating pump, and a simulation of the salt layer. By using the circulating pump, the liquid is injected into the rock powder to dissolve the rock powder in the interior. After the dissolution is basically completed, the brine is discharged by the circulating pump to realize the complete cavity construction process. The whole simulation process roughly restores the field cavity construction process, and the experimental results are closer to reality.

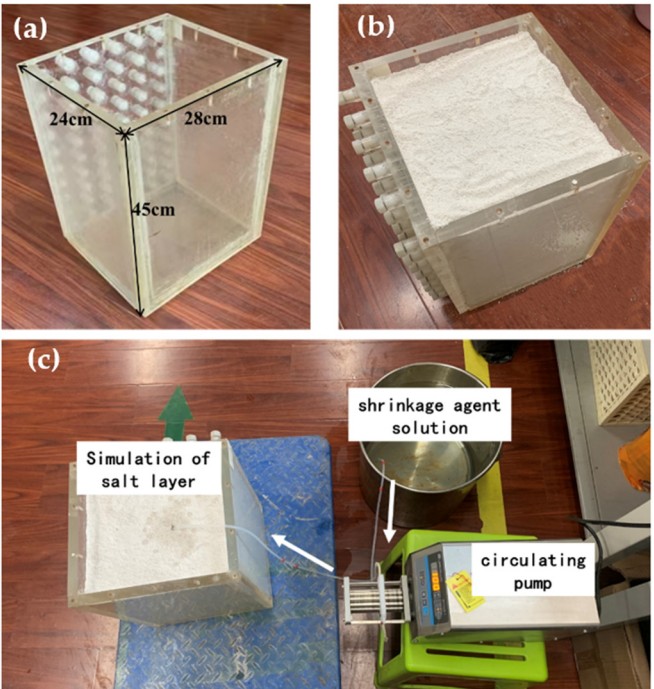

**Figure 2.** Large-scale dynamic salt solution simulation device. (**a**) A container of insoluble sediments, (**b**) a container filled with insoluble sediments, and (**c**) a large-scale dynamic salt solution simulation device.

The experimental steps are as follows: Firstly, 35 kg of rock powder was prepared according to the mass ratio of stratum insoluble matter and salt of 30:70 and then mixed evenly and poured into the experimental container in Figure 2. Then, at a constant flow rate of 27.6 mL/min, the shrinkage agent or distilled water was injected into the container where the rock powder was released and mixed evenly. The dissolution rate of the rock

powder could be judged by discharging the liquid density, and the volume V of residual solid was finally recorded. When the difference between the measured results was less than 5%, all the results were taken as the average values of the two groups. The final shrinkage is calculated by Formula (1).

*2.5. Mechanism Analysis of Shrinkage Agent*

The mechanism of the shrinkage effect of PAC30 on insoluble sediments was analyzed by an adsorption experiment, the Zeta potential, and SEM.

(1) Static adsorption of the shrinkage agent.

We added 10 mL of the 200 ppm shrinkage agent to 1.5 g of 70/100 mesh insoluble sediments [20] and let it stand at room temperature for a period of time. The experimental time was set to 20 min, 40 min, 1 h, 3 h, 6 h, and 24 h, respectively. Then, the mixture solutions were centrifuged for 15 min at 4000 RPM, and the supernatant fluids were taken for spectrophotometric measurements. From the change in absorbance over time, the loss of the shrinkage agent absorbed by insoluble sediments can be quantified.

(2) Scanning electron microscope (SEM) analysis of insoluble sediments.

The insoluble sediments after the experiment were first put into the electrical thermostat drying box and dried at 210 °C for 12 h. Then, the dried insoluble sediments were stuck onto an electron microscope metal sheet covered with conductive adhesive. To increase the electrical conductivity of the rock powder, the metal sheet with the bonded rock powder was put into a sputtered gold injector to be coated with gold. Afterward, the metal sheet coated with gold was pushed into the electron microscope cavity.

(3) Zeta potential.

The Zetasizer Nanolaser analyzer was used to measure the potential of the insoluble sediments from the gas storage and shrinkage agents. We added 140 mesh rock powers to water, stirred well, and let stand for 4 h. The supernatant was taken to test the zeta potential. This value is considered to be the potential value of insoluble sediments.

## 3. Results and Discussion

In this part, the application potential of PAC30 in salt cavern gas storage will be analyzed from the aspects of XRD analysis of insoluble sediments, the shrinkage effect, and the shrinkage mechanism of the shrinkage agent PAC30.

*3.1. XRD Analysis of Insoluble Matter in Salt Cavern Gas Storage*

Different from most of the rock salts in other countries, the extensively distributed rock salts in eastern China are all lacustrine thinly bedded types, characterized by low thickness, a high proportion of impurities, and numerous intersecting non-salt interlayers [21]. These interlayers have a wide range of thicknesses, and the content of argillaceous insoluble sediment in the interlayer ranges between 15% and 35% [22], generally accumulating a large amount of insoluble residue in the cavity after construction.

Figure 3 shows the multi-interlayer salt rock derived from well A in the Chuzhou block. It can be seen from the figure that there are obvious mudstone interlayers distributed in the salt rock. During the salt cavern leaching process, the salt in the rock is dissolved and the insoluble mudstone falls to the bottom and expands in volume. The insoluble sediments were analyzed by rock mineral composition (XRD). It can be seen from Table 1 that the formations of the insoluble matter are mainly plagioclase, quartz, clay minerals, and calcite. Table 2 is the experimental results of the clay mineral composition analysis. The main clay minerals include montmorillonite, illite, kaolinite, an illite–montmorillonite mixed layer, and a green montmorillonite mixed layer.

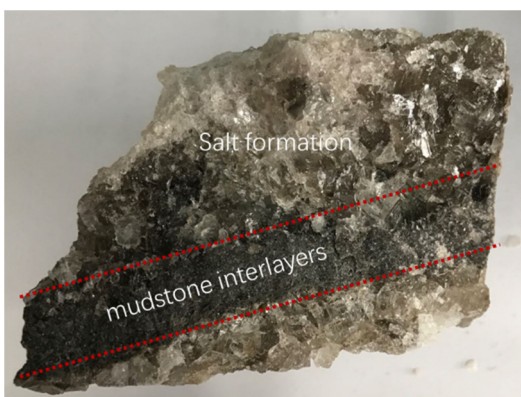

**Figure 3.** Multi-interlayer salt rock derived from well A.

**Table 1.** Mineral composition analysis of insoluble sediments in well A.

| Sample Number | Plagioclase | Quartz | Calcite | TCCM |
|---|---|---|---|---|
| Content (wt%) | 35 | 31 | 22 | 12 |

**Table 2.** Clay composition analysis of clay minerals in insoluble sediments from well A.

| Sample Number | S | It | Kao | I/S | C/S |
|---|---|---|---|---|---|
| Content (wt%) | 48 | 7 | 4 | 10 | 31 |

Note: S is montmorillonite, it is illite, Kao is kaolinite, I/S is montmorillonite mixed layer, C/S is green montmorillonite mixed layer.

### 3.2. Characterization of Shrinkage Agent PAC30

In recent years, $^{27}$Al NMR technology has been used for the determination of aluminum morphology in many fields [23–27]. Compared with other methods used for the determination of aluminum morphology, the outstanding advantage of NMR technology is that it can go deep into the substance without destroying the sample, and it will not change the chemical morphology of Al in the measured solution. What is directly obtained on the NMR spectrum of Al in the solution is a single weighted average peak for a certain morphology [28]. Therefore, $^{27}$Al NMR is a clear morphological identification technology for hydroxylated polyaluminum.

The $^{27}$Al NMR spectra of PAC30 are shown in Figure 4. δ = 80 ppm represents sodium aluminate used as the internal standard [29]. The δ values of 63 ppm and 70 ppm represent the resonance peaks of the aluminum oxygen tetrahedron in $Al_{13}$ and $Al_{30}$, respectively. As the polymerization form of $Al_{13}$ is an aggregate of 13 aluminum atoms with a Keggin structure formed by 12 hexadecylated Al octahedrons surrounding a tetrahedron with a tetrahedron through hydroxyl and oxygen bridging, only the tetrahedron with a tetrahedron at the symmetry center can produce a resonance peak at δ 63 ppm [30]. Similarly, the $Al_{30}$ polymerization form is a 30-Al aggregate formed by two δ-$Al_{13}$ polymerization forms linked by four six-coordinated Al/O octahedrons, among which only the two tetra-coordinated al atoms in the Cc symmetry can produce resonance peaks at δ 70 ppm [31].

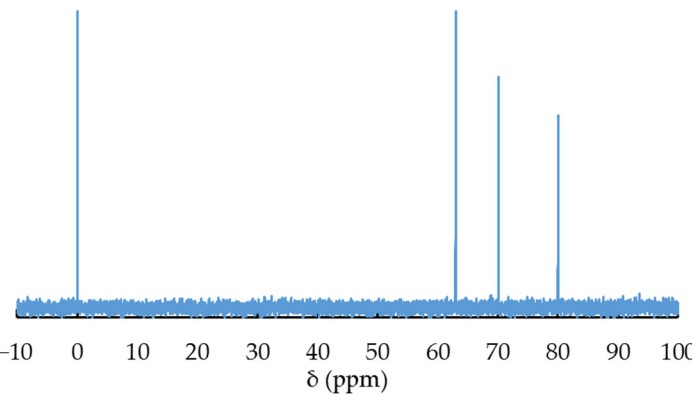

**Figure 4.** $^{27}$Al NMR spectra of PAC30.

### 3.3. Analysis of Factors Influencing the Effect of Shrinkage Agent

In the field application, the complex application environment affects the effect of the shrinkage agent. Through the shrinkage evaluation method established above, the influence of different factors on the shrinkage effect of PAC30 was studied.

#### 3.3.1. Effect of Concentrations of PAC30 on Dynamic Shrinkage

The amount of shrinkage agent is the main factor affecting the shrinkage rate. Five different concentrations of PAC30 were selected to analyze the shrinkage effect. The five concentrations were 5 ppm, 10 ppm, 20 ppm, 200 ppm, and 500 ppm. The temperature was set at 25 °C. The number of insoluble sediment mesh used in the experiment was 80–140 mesh. The experimental results are shown in Figure 5. It can be seen from Figure 5 that when the dosage of PAC30 was 20 ppm, the shrinkage rate reached 30.02%.

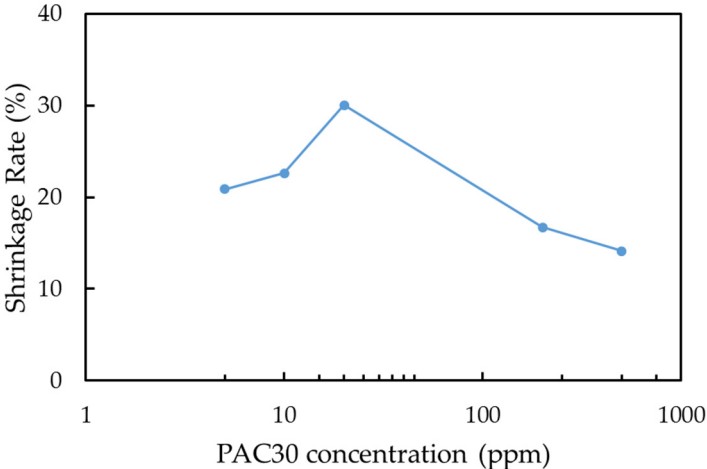

**Figure 5.** Curve of PAC30 concentration and dynamic shrinkage rate.

When the PAC30 dosage was less than 20 ppm, the shrinkage rate increased with the increase in the PAC30 concentration. When PAC30 dosage was higher than 20 ppm, the shrinkage rate decreased with the increase in the PAC30 concentration. When PAC30 is insufficient, the content of $Al_{30}$ is insufficient, so on the one hand, the charge on the surface of insoluble matter cannot be completely shielded; on the other hand, only small particle aggregates are formed, and the small aggregate particles cannot gather into large ones. Therefore, the optimum dosage of PAC30 is 20 ppm.

The addition of the shrinkage agent mainly affects the zeta potential of insoluble sediments. Due to the effect of electrostatic force, the shrinkage agent will adsorb on the surface of rock powder particles to neutralize the negative charges on the surface and reduce the expansion of insoluble sediments. At the same time, the particles can be connected by

intermolecular forces such as chain and hydrogen bonds to further compress the pore space between particles and enhance the shrinkage effect. However, when the concentration of the shrinkage agent further increases, excess positive charges will bring positive charges to the surface of the insoluble sediments, causing the electrostatic repulsion to increase and the effect of the shrinkage agent to decrease. This explains why, after reaching the peak, the shrinkage effect decreases [32].

In the process of the experiment, continuous injection of the PAC30 solution will result in more of the shrinkage agent in the insoluble sediments, so less shrinkage agent is needed to achieve the optimal volumetric shrinkage. The PAC30 solution with a 0.002% concentration was used for the dynamic shrinking test in subsequent experiments.

### 3.3.2. Effect of Temperature on Shrinkage Effect of PAC30

Under formation conditions, the temperature is one of the important factors affecting liquid performance. Figure 6 compares the shrinkage effects of PAC30 at 25 °C and 60 °C. It can be seen from Figure 6 that the shrinkage rate gradually increased with the concentration at two temperatures. The expansion rate at 60 °C is significantly lower than that at 25 °C. This is because when the temperature increases, the Brownian motion of PAC30 can accelerate, which speeds up the interaction between the PAC30 molecules and the upper rock particles. So, the shrinkage rate was accelerated at the beginning of 60 °C. However, when the volume of upper rock particles was compacted (as shown in Figure 7), it further hindered the diffusion of PAC30 at this depth, which resulted in the reduction of shrinkage. Moreover, when the temperature is too high, the hydrolysis speed of PAC30 is too fast, leading to the aging, shrinkage, and fracture of flocculant molecules, and finally, to the continuous decline of the flocculation capacity of PAC30 [33].

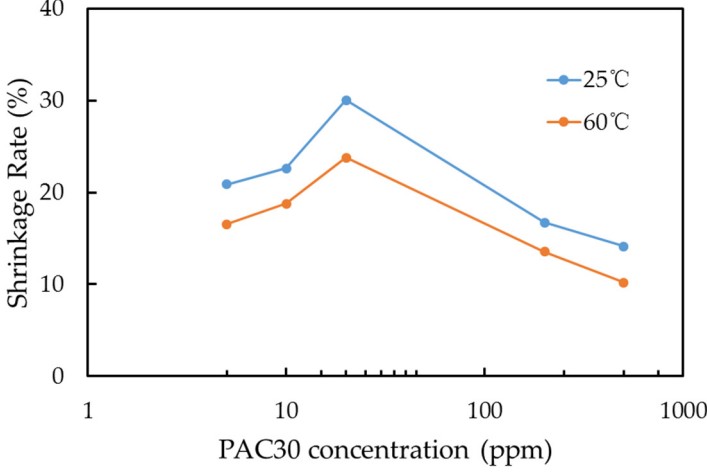

**Figure 6.** Curve of expansion rate of PAC30 with temperature and concentration.

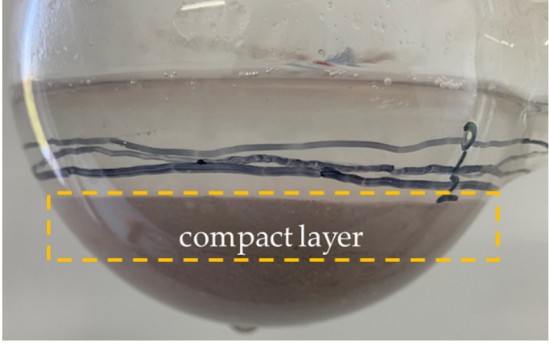

**Figure 7.** Compact layer formed after injection of PAC30 solution.

### 3.3.3. Effect of Particle Size of Insoluble Sediments on Shrinkage Agent PAC30

The mesh size of insoluble sediments used to evaluate the dynamic shrinkage effect of PAC30 is 80–140 mesh, and the shrinkage rate is 30.02%. However, the particles of insoluble sediments from the gas storage have a wide size range. In order to evaluate the corresponding relationship between the shrinkage agent and the particle size of insoluble sediments, the particle size composition of two wells was simulated based on field sampling results [3,21].

Table 3 shows the particle size distribution data of this experiment. Particle size combination 1 designed in the experiment was significantly smaller than that of particle size combination 2. The results of the dynamic dissolution experiment are shown in Figure 8. It is concluded that the smaller the particle size of the rock powder is, the larger the shrinkage rate is and the better the shrinkage effect is. It is proposed that there are two main reasons: First, the smaller the particle size, the larger the specific surface, and the larger the area between the particle and the shrinking agent. It resulted in stronger adsorption and compaction. Secondly, it can be seen in Figure 9 that there are many small particles suspended on the liquid surface. They can be taken away with the flow of brine, which is consistent with the literature results. Sun found that the particles with a size less than 0.1 mm do not settle in brine and are discharged with the transport of brine in the cavity [33–35]. With the discharge of the parts of small particles, the space occupied by insoluble sediments at the bottom of the salt cavern is reduced to a certain extent, so that the space of the gas storage is further enlarged.

**Table 3.** The particle size distribution combination used in the experiment.

| Particle Size (mesh) | 10~14 | 14~24 | 24~32 | 32~60 | 60~80 | 80~140 |
|---|---|---|---|---|---|---|
| particle size combination 1 (%) | 70.0 | 16.0 | 4.0 | 4.0 | 4.0 | 2.0 |
| particle size combination 2 (%) | 88.0 | 3.4 | 4.0 | 1.6 | 2.0 | 1.0 |

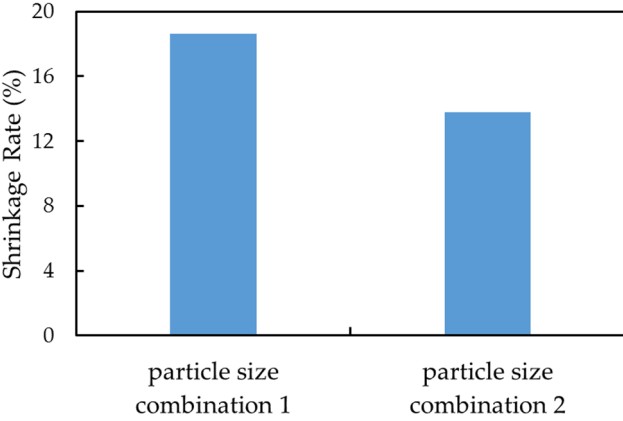

**Figure 8.** Relationship between shrinkage rate and particle size combination of PAC30.

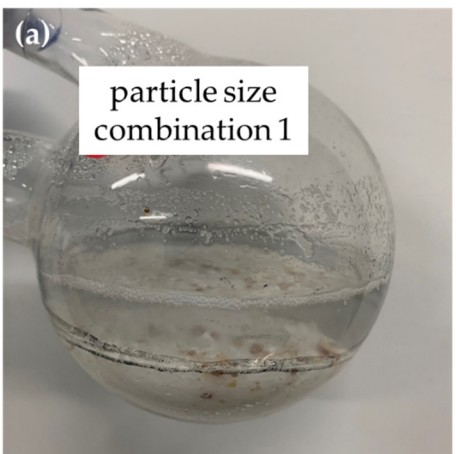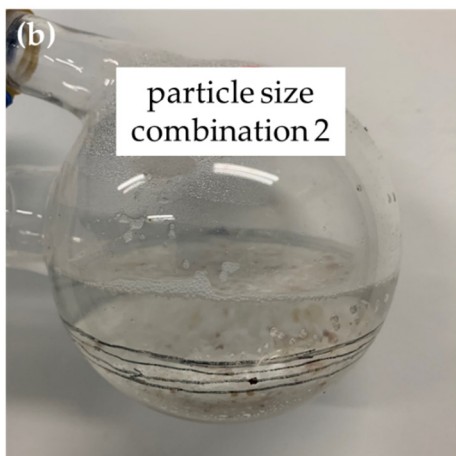

**Figure 9.** Experimental diagram of particle size dynamic shrinkage. (**a**) The experimental diagram of particle size combination 1, (**b**) the experimental diagram of particle size combination 2.

3.3.4. Physics Simulation Experiment with Large Size Device

To get closer to the cavern construction process on site, a large experimental device was developed, as shown in Figure 2.

First, 35 kg of rock powder was prepared according to the mass ratio of 30:70 of insoluble sediments and stone salt. Then it was mixed, stirred evenly, and poured into the experimental container. Again, at a constant flow rate of 27.6 mL/min, the shrinkage agent solution or distilled water was injected into the container. Whether salt rock was completely dissolved could be judged by the density of the discharged liquid, and the final volume of the remaining solid was recorded. The whole dissolution time of each experiment was approximately 10 days. The volume of mixed rock powders in this experiment was 26,880 cm$^3$. The residual volume after the injection of tap water was 10,214.4 cm$^3$ and accounts for 38% of the total volume. After the injection of the 20 ppm PAC30 solution, the volume of the accumulation was reduced to 8064 cm$^3$, and the volume was reduced by 21.05% compared with that of tap water. If the volume of insoluble sediments accounted for 40% of the total volume of the gas storage, it can be calculated that the storage volume increased by 8% after the injection of the 20 ppm PAC30 solution. The results of this experiment directly proved that the shrinkage effect of PAC30 is excellent. PAC30 can effectively reduce the volume of insoluble sediments and increase the storage volume of a bedded salt cavern.

*3.4. Study on Mechanism of Shrinkage*

Due to the negative charge on the surface of the rock powder particles after hydration, the surrounding ions with opposite charges are attracted, and these opposite charges are distributed in a diffusion state at the two-phase interface to form a diffusion double layer. The surface of the rock powder is closely connected to water molecules connected by hydrogen bonds and some cations with hydration shells, which together constitute the adsorption solvation layer, which is macroscopically reflected as volume expansion. The main function of the shrinkage agent is to adsorb the surface of the rock powder, neutralize the charge, aggregate the rock powder, and reduce the space occupied by the rock powder. The adsorption of PAC30 on different mineral surfaces was studied in this paper.

3.4.1. Adsorption of PAC30 on Mineral Surface

According to the experimental parameters given in Section 2.4, the total injection amount of PAC30 was 40.32 mg and the ratio of PAC30 to insoluble sediments was 0.672 mg/g (PAC30/insoluble sediments) in 168 h. The adsorption experiment should ensure that the content of PAC30 in the solution is higher than the adsorption amount on the insoluble surface. The amount of insoluble sediments used in the adsorption experiment

is 1.5 g, and at least 1mg of PAC30 is required. Therefore, 200 ppm was selected as the concentration used in the adsorption experiment.

The static adsorption capacities of rock powder at different times in the adsorption process were measured. The experimental results are shown in Figure 10, and the adsorption capacities of different minerals to PAC30 are shown in Table 4.

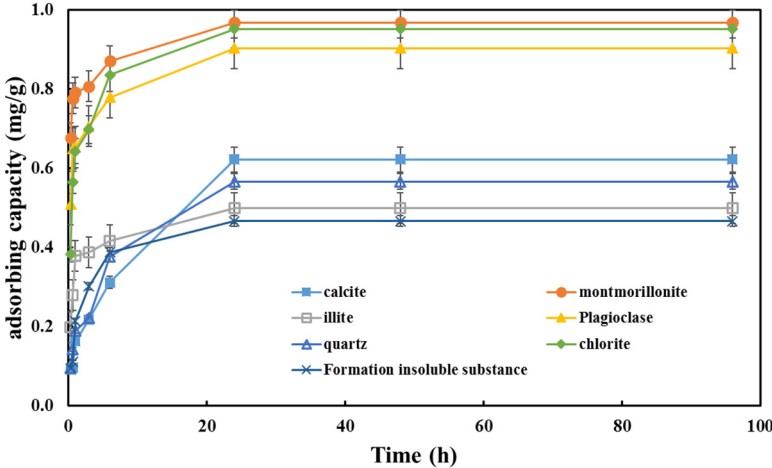

**Figure 10.** Adsorption curves of PAC30 in different minerals.

**Table 4.** Adsorption capacity of different minerals on PAC30 and charge of mineral surface.

| Rock Powder Minerals | Montmorillonite | Quartz | Plagioclase | Illite | Calcite | Chlorite | Insoluble Sediments |
|---|---|---|---|---|---|---|---|
| Zeta potential (mV) | −37 | −4.43 | −17.1 | −8.48 | −1.96 | −18.8 | −18.1 |
| adsorption (mg/g) | 0.967 | 0.566 | 0.902 | 0.499 | 0.622 | 0.951 | 0.466 |

The adsorption capacity is a measure of the strength of mutual repulsion or attraction between shrinkage agents and minerals. A large adsorption capacity indicates a strong interaction between the shrinkage agent and the mineral, and a small adsorption capacity indicates a weak interaction between the shrinkage agent and the mineral [36]. Only when the force between the shrinkage agent and the mineral is strong can the effect of shrinkage be achieved.

It can be seen from Table 3 that the PAC30 has a larger adsorption capacity on the surface of clay minerals than on the surfaces of other minerals. The adsorption capacity of PAC30 is proportional to the surface charge of the mineral. For example, montmorillonite is easily hydrated and dispersed in water, with a surface charge of −37 mV. PAC30, with positive charges on the surface, can form a strong electrostatic force with montmorillonite, and has an adsorption capacity of 0.967 mg/g on the surface of montmorillonite. Its adsorption capacity on the surface of the insoluble sediments was 0.466 mg/g, indicating that PAC30 can fully interact with the insoluble sediments. Moreover, the adsorption of PAC30 on the mineral surface became stable 24 h into the experiments; with the further extension of the adsorption time, its adsorption capacity did not change significantly.

### 3.4.2. Zeta Potential

The Zeta potential is an important characteristic parameter of the diffusion layer, and its value can reflect the thickness of the diffusion layer [37–39]. The lower the absolute value of the Zeta potential is, the fewer adsorbed cations in the diffusion layer, and the thinner the diffusion layer is. Conversely, the higher the absolute value of the Zeta potential, the more adsorbed cations there are and the thicker the diffusion layer is.

In order to study the Zeta potential changes after the interaction between different concentrations of PAC30 and insoluble sediments, according to the adsorption capacity of PAC30 on the insoluble sediment's surface, PAC30 with a concentration of 100 ppm to 500 ppm was mixed with 15 g of insoluble sediments, and its Zeta potential was tested.

It can be seen from Table 5 that with the increase in the PAC30 concentration, Zeta potential gradually turns from negative to positive, indicating that when the concentration of PAC30 added is lower than 200 ppm, the thickness of the diffusion double layer decreases, the interparticle repulsion decreases, and the insoluble sediment particles aggregate. When the concentration of PAC 30 added exceeds 200 ppm, the concentration of positive charges between particles increases and particles begin to disperse again. In the process of decline of potential absolute values, PAC30 can compress the electric double layer by electric neutralization to cause the water film combined with clay particles to become thinner and the rock powder particles aggregate with each other, achieving the purpose of shrinkage of insoluble sediments. However, when the added cations exceed a certain concentration, the surfaces of clay particles are positively charged, which, in turn, attract negatively charged anions and form new diffusion double layers, so the repulsion between clay particles begins to increase again, resulting in an increase in the diffusion layer thickness.

**Table 5.** Zeta potential values of insoluble sediments after adding different concentrations of PAC30 solution.

| Insoluble Sediments Amount (g) | PAC30 Concentration (ppm) | Zeta Potential (mV) |
|---|---|---|
| | 0 | −18.10 |
| | 100 | −11.40 |
| 15 | 200 | 0.36 |
| | 300 | 4.73 |
| | 500 | 14.90 |

### 3.4.3. SEM Analysis

In order to further explore the shrinkage mechanisms of the shrinkage agent, the microstructure of the underlying insoluble sediments was observed by scanning electron microscopy to study the insoluble sediment treated with the shrinkage agent.

Figure 11a is an SEM image of the insoluble sediments. It can be seen from this figure that the particles were in loose accumulation, with no aggregation. Figure 11b shows the SEM image of the insoluble sediments after being treated by PAC30. The PAC30 with a molecular size of more than 5 nm can adsorb at multiple points onto rock particles, causing particles of expanded insoluble sediments to agglomerate into a larger whole, thereby reducing the spacing between the particles. The smaller the floc size, the higher the shrinkage efficiency is.

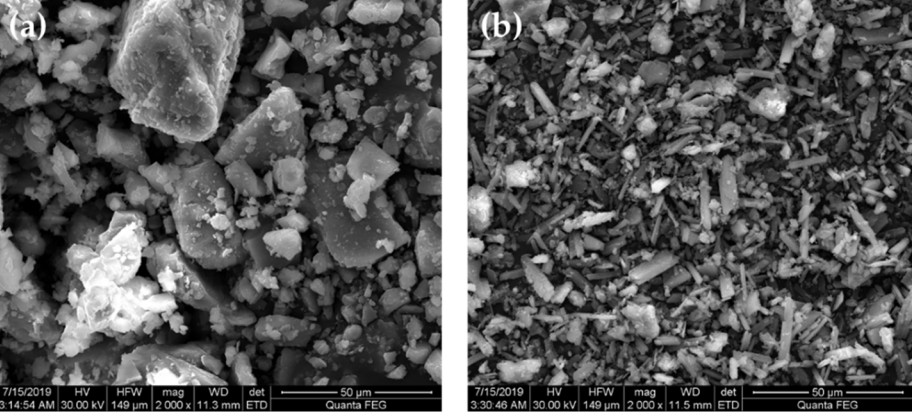

**Figure 11.** SEM images of insoluble sediments. (**a**) The insoluble sediments; (**b**) the insoluble sediments treated with PAC30.

The basic principle of the shrinkage agent is to compress the double electric layer via bridge multipoint adsorption to achieve volume compression. Inorganic polymer shrinkage agents, with large molecular sizes, can absorb on the surface of insoluble sediments to neutralize negative charges, and cause insoluble sediments particles to become more compacted

through bridging and multi-point adsorption, as shown Figure 12. PAC30 is one such material, with a Keggin structure and a molecular weight between 2000 and 4000 g/mol, which can form multi-point adsorption and bridges between rock powder particles.

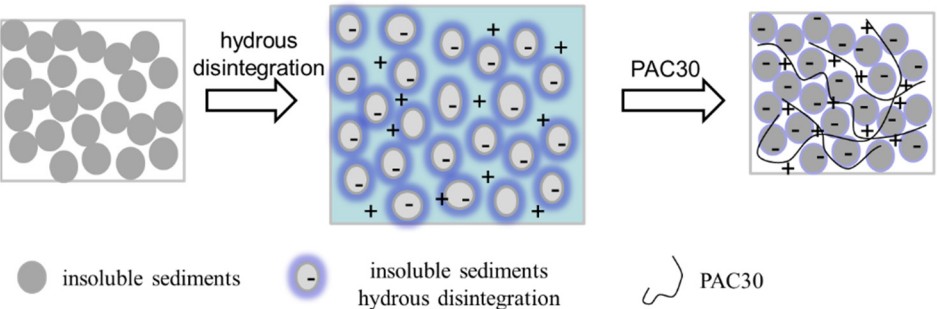

**Figure 12.** Schematic diagram of action mechanism of shrinkage agent.

## 4. Conclusions

In order to reduce the volumetric expansion of the insoluble sediments at the bottom of the salt cavern gas storage, a chemical shrinkage agent, PAC30, was designed and synthesized. A dynamic shrinkage experimental apparatus was established to study the effects of temperature, particle size, and concentration on the shrinkage efficiency of PAC30. Furthermore, its shrinkage mechanism was clarified by SEM and Zeta potential tests. The main conclusions are as follows:

Firstly, the self-synthesized PAC30 is an effective chemical shrinkage agent, especially when the concentration is 20 ppm. The result of the NMR experiment shows that it has a Keggin structure and forms a 30-Al polymer consisting of two δ-Al13 structures. Secondly, the shrinkage efficiency is mainly controlled by adsorption and electrostatic interactions. When the Zeta potential of the insoluble sediments after PAC30 treatment was close to 0 mv, the volume of the insoluble sediments was the smallest. Through SEM observations, insoluble sediments accumulated more closely after adding PAC30. The main shrinkage mechanism is that PAC30, with a molecular weight between 2000 and 4000 g/mol, can absorb on the surface of insoluble sediments to neutralize negative charges and cause insoluble sediments particles to become more compacted through bridging and multi-point adsorption, which is conducive to reducing the expansion volume. Finally, considering the field conditions, such as temperature and particle size, the shrinkage rate of PAC30 was intensively evaluated. A large dynamic shrinkage device was built, and the maximum dynamic shrinkage rate of PAC30 was up to 21.05%. With the increase in insoluble sediments' particle size, the shrinkage efficiency of PAC30 decreased rapidly. This is because large particles have a smaller specific surface area. A high temperature accelerated the appearance of the compact layer and reduced the shrinkage rate. The adsorption capacity of PAC30 on the insoluble sediments' surface was 0.466 mg/g at the optimal dosage. The experimental results of this paper provide a method to expand the capacity of salt cavern gas storage, and understanding the shrinkage mechanism will guide the design of new chemical shrinkage agents.

**Author Contributions:** Conceptualization, F.Z.; methodology, B.W. and M.Z.; validation, W.D., Q.W. and J.Q.; formal analysis, W.L.; investigation, T.L.; resources, F.Z.; data curation, Y.L.; writing—original draft preparation, Y.L.; writing—review and editing, Y.L. and T.L.; visualization, B.W.; supervision, F.Z. and J.Q. All authors have read and agreed to the published version of the manuscript.

**Funding:** This work was financially supported by the Strategic Cooperation Technology Projects of CNPC and CUPB (ZLZX2020-01).

**Conflicts of Interest:** The authors declare no conflict of interest.

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
