# Peer review of "Study and Mechanism Analysis on Dynamic Shrinkage of Bottom Sediments in Salt Cavern Gas Storage"

_processes, doi:10.3390/pr10081511_

Round 1

Reviewer 1 Report

Highlight changes in yellow in a next revision, please. No track changes.

I believe language needs to be carefully revised. Do not use work but study. And do not refer to ourselves or we, etc.

In this work, 15
a method reducing the volume of argillaceous insoluble sediments with particle sizes ranging from 16
30 mesh to 140 mesh by chemical shrinkage agent has been proposed. Firstly, the inorganic polymer 17
shrinkage agent PAC30 was synthesized by ourselves,

All abbreviations need to be defined at first use.

Finally, by 20
means of SEM,

See that sentences need to be complete. Optimum concentration of what?

The experimental 22
results show that the optimum concentration for dynamic shrinkage and expansion experiment is 23
0.002%.

And again, with different values.

“The experimental 77

results show that the shrinkage agent has the best shrinkage effect at the concentration of 78

0.02 %, and can expand the effective volume of gas storage.”

Shrinkage of what?

Formation temperature and particle size of rock powder are the main factors affecting the 24
effect of shrinkage.

And many more

This is an example of an unclear sentence.

The field application potential of PAC30 was verified by large dynamic shrink- 25
age experiment.

 I would like to see more quantitative data in abstract.

Please address all proper subscripts in chemical formulas.

(e.g., natural gas, oil, H2 etc.)”

I do not understand nor the sentence nor the inclusion of only one reference.

“To sum up, most of the existing studies focused on the accumulation mode [12],“

Surely they were purchased. It would be more important to add data on the quality.

“2.1. Materials 81

Sodium chloride (NaCl), aluminum chloride (AlCl3), potassium hydroxide (KOH) 82

were purchased from China National Pharmaceutical Group. The insoluble sediments 83

were taken from a gas storage in Chuzhou, China.”

Please do not list please avoid so many headings.

“2.4. Mechanism analysis of shrinkage agent 110

2.4.1 Static adsorption of shrinkage agent 111

(1)”

There is absolutely no interest in including a subheading with such tiny text.

“2.4.3 zeta potential 126

Zetasizer Nanolaser analyzer was used to measure the potential of the insoluble sed- 127

iments from the gas storage and the shrinkage agents.”

This is extremely unclear.

2.5. Establish a shrinkage agent evaluation method

Again, please DO NOT list…

The equations, not formulas are usually based in known data. Please at the necessary reference data immediately before the equation is presented.

If that is not the case, then explain it. Also, address every italics in variables.

⑤ The calculation formula of shrinkage rate is:

Is image has no interest at all. The quality is terrible. The equipment cannot be seen. It is not properly identified, and the legend says nothing.

“Figure.1 Apparatus for dynamic shrinkage experiment”

Please do not use Red Letter is insulting in specific countries. Again, the picture has no quality and no interest as presented. Also, if it is a grouped set of figures, each one should have a letter in the different subscription after the main caption by letter.

Figure.2 Large-scale Dynamic Salt Solution Simulation Device

Not sure if the authors are used to the publishing process because we cannot have a figure immediately after the subheading.

3. Results and discussion 188
3.1. XRD analysis of insoluble matter in salt cavern gas storage

Again the caption is unclear. Analysis of what at what level?

Table 1 Analysis of insoluble sediments

See that unit should be inside ().

Figure.5 Effect of standing time on shrinkage

Again, the caption is not clear to me. The effect word should not even be there because that is to be discussed during the manuscript.

And again. Concentrations of what?

3.3.2 Effect of concentrations on dynamic shrinkage

And again please compare with values previously presented.

“The maximum shrinkage rate was reached when the concentra- 258

tion of the four shrinkage agents was 0.002%,”

“PAC30 solution with 0.002% concentration was used for dynamic shrinking test in subse- 264

quent experiments”

The influence could be the title of a hitting not of a caption. See that again. Graphics have very different aspects. This one for example is really outdated. There are ways to make things interesting. And again, shrinkage of what?

Figure.7 Influence of temperature on dynamic shrinkage                                                                                            

What is the interest of including Chinese characters to an international audience? See again dot. It is not interesting at all. This figure has no value, no interest, and it is grouped.

Figure 9. Experimental diagram of particle size dynamic shrinkage.

Please compare the aspect and colours of this graphic with other ones. There should be a coherence.

Figure.10 Adsorption curves of PAC30 in different minerals

Again, units inside brackets.

PAC30 concentration /%

The captions need to be hugely improved. The reader has to get nothing at all, so there are many things to change in this manuscript. Remove the upper letters in bold and see that they serve no purpose since they are not identified in the subscriptions. No upper letters, but lower letters and no red.

Figure.11 SEM images of insoluble sediments

Every caption must be made self explanatory. They are not. Unless I’m wrong, the circles are distorted.

Figure.12 Schematic diagram of action mechanism of shrinkage agent

Conclusion section must obey to a specific structure that is. Brief contextualization, an methodology, main findings in practical implications, limitations and further prospects if that is the case.

This idea of listing everything does not serve well the purpose of knowledge because it forces the reader to conclude. Everything needs to be related, so every content needs to be interconnected. So remove the list.

“The following understandings and conclusions have 415

been reached: 41”

Please carefully cheque all values presented during the text. Please respect the spacing before units cheque international unit system.

15g insoluble sediments treated with 0.02%

I hope the authors understand that this comment aimed to assist the authors in improving the manuscript. There is plenty of room for improvement to achieve a relevant manuscript when the reviewer needs to be focused on minor corrections like the formal aspect. It cannot focus on the essential ones. Also, please compare the values being presented here with the lack of values in the abstract.

Authors need to understand that if the article is aimed at an international indexed journal, then references must be made international too, including authors outside the regional setting.

The list of references is extremely scarce.

It needs to include much more references. This is not a comment…

Reviewer 2 Report

1) "The calculation formula of the final inflation rate is:" (the row 165) ???
2) "The experimental device as shown in Figure 3 was self-built device, which is a cuboid container of 24cm×28cm×40cm for storing rock powder." (rows 169-170). It should sound like this: "The experimental device as shown in Figure 2 was a self-built device, which is a cuboid container of 24cm×28cm×40cm for storing rock powder."
3) Figure 2 requires discussion and presentation.
4) The proportions in Table 1 are wrong. The sum of the percentages exceeds 100% resulting in a total composition of 105%.
5) Figure 4 requires a more detailed discussion.
6) Table 3 and Figures 8 and 9 require further discussion.
7) Personally, I was not able to identify the last sent, to reference 21.

Round 2

Reviewer 1 Report

Highlight changes in yellow in a next revision, please. No track changes.

Language needs additional clarification. Everything authors write in a scientific manuscript must be meaningful, must be significant.

“To sum up, most of the existing studies ”

Existing studies on what? See that this is a different sentence and a new sentence.

Is this a new equation? Not based in known data. Is it completely original? No. So please add the necessary citation immediately. The equation is presented.

“tion formula of shrinkage rate is:”

Also see that italics in parameters were not respected in the text.

Of course, this is an apparatus, Why add that word in the caption…

“Figure.1 Apparatus for dynamic shrinkage experiment”

Mentioned before… and ignored…

“Q14: Is image has no interest at all. The quality is terrible. The equipment cannot be seen. It is not properly identified, and the legend says nothing. “Figure.1 Apparatus for dynamic shrinkage experiment””

Please do not use upper letter in letters identifying the figures.

Authors used to the publishing process. No, this is not possible. We cannot have a list immediately after a heading. Lists must be avoided at all costs because the content needs to be interconnected. It’s very easy to make a list. It’s difficult to connect the content.

“.5. Mechanism analysis of shrinkage agent 165

(1) Static adsorption of shrinkage agent”

(2)

Again, I do not find the captions to be self-explanatory at all.

“Figure.3 Formation coring of Well A in Chuzhou”

Please do not use abbreviations in captions. The reader cannot be forced to go back and check what is it about…

“Table 1 XRD analysis of insoluble sediments in well A”

Well, I have never seen such a graphic see that again as an as mentioned in my previous revision, units must be included inside brackets. Every axis must have a legend where is the legend of YY axis…

“Figure.4 27Al NMR spectra of PAC30 “

Again, I must insist. Captions are just not explanatory. Dissolution of what? See that the reader does not have to go back to the text and see what are authors talking about.

“Figure.7 Compact layer formed during dissolution”

I have mentioned this before. This is a group of two figures, so they need to be identified by a separate letter and a separate sub caption must be included after the main caption.

“Figure 9. Experimental diagram of particle size dynamic shrinkage”

I will repeat what I said earlier.

“Q26: Conclusion section must obey to a specific structure that is. Brief contextualization, an methodology, main findings in practical implications, limitations and further prospects if that is the case. This idea of listing everything does not serve well the purpose of knowledge because it forces the reader to conclude. Everything needs to be related, so every content needs to be interconnected.”

Please completely revise the conclusions section. I would advise the authors to read some significant papers.

Again, I hope the authors are able to understand that the corrections above will contribute to a much more relevant paper. Able to be cited.

The number of references in the manuscript is still very scarce. Authors again would need to include more international authors.

Again mentioned previously…

“Authors need to understand that if the article is aimed at an international indexed journal, then references must be made international too, including authors outside the regional setting. The list of references is extremely scarce. It needs to include much more references. This is not a comment…”

Round 3

Reviewer 1 Report

Highlight changes in yellow in a next revision, please. No track changes.

Well, again, I specifically asked no track changes and manuscript cannot be properly reviewed if previous changes and new changes are there.

Well, I do not have to agree with it and to me it makes no sense to have parameters in non italics.  All my papers have parameters in italics.

Thank you for your valuable suggestions. The formula adds references. The format of the parameters follows the journal template, so they are not set to italics.

And again authors seemed to list everything. Not clear what is knew and what is not.

“Step3: The metal sheet coated with gold was pushed into the electron microscope 233

cavity. 234

(3) zeta potential”

Authors ignore my comments relating figure four. The Y axis needs to have a legend.

“I will repeat what I said earlier.

Q26: Conclusion section must obey to a specific structure that is. Brief contextualization, an methodology, main findings in practical implications, limitations and further prospects if that is the case. This idea of listing everything does not serve well the purpose of knowledge because it forces the reader to conclude. Everything needs to be related, so every content needs to be interconnected.

Please completely revise the conclusions section. I would advise the authors to read some significant papers.”

I cannot  accept the listing in the conclusions section. Please revise as previously advised. There cannot be no list.

Round 4

Reviewer 1 Report

I will repeat what I said earlier:

Q26: Conclusion section must obey to a specific structure that is. Brief contextualization, an methodology, main findings in practical implications, limitations and further prospects if that is the case. This idea of listing everything does not serve well the purpose of knowledge because it forces the reader to conclude. Everything needs to be related, so every content needs to be interconnected.

Please completely revise the conclusions section. I would advise the authors to read some significant papers.”

There is no contextualization, there are no implications, for example, additionally, the article continues to have extensive lists. A references is nor added along the equation number, never seen that. The authors need to be willing to change.
